# Metabolic risk factors and incident advanced liver disease in non-alcoholic fatty liver disease (NAFLD): A systematic review and meta-analysis of population-based observational studies

Helen Jarvis[1]*, Dawn Craig[1], Robert Barker[1], Gemma Spiers[1], Daniel Stow[1], Quentin M. Anstee[2,3‡], Barbara Hanratty[1‡]

**1** Population Health Sciences Institute, Faculty of Medical Sciences, Newcastle University, Newcastle upon Tyne, United Kingdom, **2** Translational and Clinical Research Institute, Faculty of Medical Sciences, Newcastle University, Newcastle upon Tyne, United Kingdom, **3** NIHR Newcastle Biomedical Research Centre, Newcastle upon Tyne Hospitals NHS Trust, Newcastle upon Tyne, United Kingdom

‡ These authors are joint senior authors on this work.
* helen.jarvis2@newcastle.ac.uk

**Data Availability Statement:** All relevant data are within the manuscript and its supporting information files.

## Abstract

### Background

Non-alcoholic fatty liver disease (NAFLD) is a leading cause of chronic liver disease worldwide. Many individuals have risk factors associated with NAFLD, but the majority do not develop advanced liver disease: cirrhosis, hepatic decompensation, or hepatocellular carcinoma. Identifying people at high risk of experiencing these complications is important in order to prevent disease progression. This review synthesises the evidence on metabolic risk factors and their potential to predict liver disease outcomes in the general population at risk of NAFLD or with diagnosed NAFLD.

### Methods and findings

We conducted a systematic review and meta-analysis of population-based cohort studies. Databases (including MEDLINE, EMBASE, the Cochrane Library, and ClinicalTrials.gov) were searched up to 9 January 2020. Studies were included that reported severe liver disease outcomes (defined as liver cirrhosis, complications of cirrhosis, or liver-related death) or advanced fibrosis/non-alcoholic steatohepatitis (NASH) in adult individuals with metabolic risk factors, compared with individuals with no metabolic risk factors. Cohorts selected on the basis of a clinically indicated liver biopsy were excluded to better reflect general population risk. Risk of bias was assessed using the QUIPS tool. The results of similar studies were pooled, and overall estimates of hazard ratio (HR) were obtained using random-effects meta-analyses. Of 7,300 unique citations, 22 studies met the inclusion criteria and were of sufficient quality, with 18 studies contributing data suitable for pooling in 2 random-effects meta-analyses. Type 2 diabetes mellitus (T2DM) was associated with an increased risk of

**Funding:** The funding for this review was provided by an NIHR In Practice Fellowship award IPF-2018-12-008 (HJ). National Institute of Health Research (NIHR) website: https://www.nihr.ac.uk/ The funders had no role in study design, data collection and analysis, decision to publish, or preparation of the manuscript.

**Competing interests:** I have read the journal's policy and the authors of this manuscript have the following competing interests: QMA is coordinator of the LITMUS (Liver Investigation: Testing Biomarker Utility in Steatohepatitis) consortium funded by the IMI2 H2020 Framework Program of the European Union under Grant Agreement 777377; has been a consultant for 89Bio, Abbott Laboratories, Acuitas Medical, Allergan/Tobira, Altimmune, AstraZeneca, Axcella, Blade, BMS, BNN Cardio, Celgene, Cirius, CymaBay, EcoR1, E3Bio, Eli Lilly & Company Ltd., Galmed, Genentech, Genfit SA, Gilead, Grunthal, HistoIndex, Indalo, Imperial Innovations, Intercept Pharma Europe Ltd., Inventiva, IQVIA, Janssen, Madrigal, MedImmune, Metacrine, NewGene, NGMBio, North Sea Therapeutics, Novartis, Novo Nordisk A/S, Pfizer Ltd., Poxel, ProSciento, Raptor Pharma, Servier, Terns, Viking Therapeutics; and has received research support from Abbvie, Allergan/Tobira, AstraZeneca, GlaxoSmithKline, Glympse Bio, Novartis Pharma AG, Pfizer Ltd., Vertex. The other authors have declared that no competing interests exist.

**Abbreviations:** HDL, high-density lipoprotein; HR, hazard ratio; NAFLD, non-alcoholic fatty liver disease; NASH, non-alcoholic steatohepatitis; NCEP ATP III, National Cholesterol Education Program Adult Treatment Panel III; NICE, National Institute for Health and Care Excellence; T2DM, type 2 diabetes mellitus; WC, waist circumference; WHR, waist-to-hip ratio.

incident severe liver disease events (adjusted HR 2.25, 95% CI 1.83–2.76, $p < 0.001$, $I^2$ 99%). T2DM data were from 12 studies, with 22.8 million individuals followed up for a median of 10 years (IQR 6.4 to 16.9) experiencing 72,792 liver events. Fourteen studies were included in the meta-analysis of obesity (BMI > 30 kg/m$^2$) as a prognostic factor, providing data on 19.3 million individuals followed up for a median of 13.8 years (IQR 9.0 to 19.8) experiencing 49,541 liver events. Obesity was associated with a modest increase in risk of incident severe liver disease outcomes (adjusted HR 1.20, 95% CI 1.12–1.28, $p < 0.001$, $I^2$ 87%). There was also evidence to suggest that lipid abnormalities (low high-density lipoprotein and high triglycerides) and hypertension were both independently associated with incident severe liver disease. Significant study heterogeneity observed in the meta-analyses and possible under-publishing of smaller negative studies are acknowledged to be limitations, as well as the potential effect of competing risks on outcome.

## Conclusions

In this review, we observed that T2DM is associated with a greater than 2-fold increase in the risk of developing severe liver disease. As the incidence of diabetes and obesity continue to rise, using these findings to improve case finding for people at high risk of liver disease will allow for effective management to help address the increasing morbidity and mortality from liver disease.

## Trial registration

PROSPERO CRD42018115459.

## Author summary

### Why was this study done?

- This review gathered together the existing evidence on which metabolic risk factors are most associated with severe forms of liver disease.

- Many people have risk factors for developing fat on their livers, but most will not develop severe liver disease.

- Knowing which individuals are at greatest risk of liver disease will facilitate targeting of interventions to people with the greatest potential to benefit.

### What did the researchers do and find?

- Combining the results of many individual studies, we found that type 2 diabetes was associated with a more than 2-fold increase in the likelihood of developing severe liver disease.

- Other metabolic risk factors (obesity, fat levels in the blood, and high blood pressure) were also reviewed. Obesity was also associated with an increased risk of liver disease, but to a lesser extent than type 2 diabetes.

- There was less information available on the other risk factors.

**What do these findings mean?**

- These findings mean that when health professionals are trying to find people at high risk of significant metabolic liver disease, they should focus on those who already have diabetes.

- The relative lack of evidence on the effects of other metabolic risk factors and combinations of these risk factors in predicting liver disease should be a focus of research in the future.

## Introduction

Non-alcoholic fatty liver disease (NAFLD) is a leading cause of chronic liver disease worldwide, with an estimated population prevalence rate of up to 30% in Europe [1]. Progressive liver disease is asymptomatic and usually diagnosed late, at the stage of decompensated cirrhosis, when intervention is less effective and mortality rates are high. Most people with NAFLD will not develop progressive disease (advanced fibrosis/cirrhosis), but recent guidelines have stressed the importance of identifying the minority that will [2]. Data from biopsy studies have shown that the histological staging of liver fibrosis is one of the most important prognostic factors in NAFLD. Advanced fibrosis is associated with severe liver-related outcomes and increased mortality [3,4]. Without undertaking a biopsy, advanced liver disease can be ruled out with acceptable accuracy using non-invasive biomarkers or simple clinical scores [5,6]. However, validation of these methods in unselected populations is limited. Furthermore, in settings where the pretest probability of advanced fibrosis is low, the positive predictive value of non-invasive tests will fall and lead to many false positives [7]. This highlights a need to clearly define the at-risk population before employing these tests.

Case finding for advanced liver disease amongst adults with type 2 diabetes mellitus (T2DM) or metabolic syndrome is recommended by the European Association for the Study of the Liver (for those aged over 50 years) and the American Diabetes Association [2,8]. Ongoing studies are providing evidence for the clinical effectiveness and cost-effectiveness of risk-factor-based case finding for NAFLD in unselected populations [9–11]. However, the high and rising prevalence of risk factors for NAFLD means that the introduction of such programmes at scale will be costly, and neither the UK National Institute for Health and Care Excellence (NICE) nor the American Association for the Study of Liver Diseases has recommended case finding in primary care in their latest NAFLD guidelines [12,13]. In the absence of proactive case finding and assessment of high-risk individuals, case ascertainment is inconsistent and largely opportunistic, based on chance findings of abnormal blood tests or imaging carried out for other purposes. The current approach will not identify those at most risk. In many care settings, this means diagnosis late in the disease natural history, with limited scope for effective intervention [14].

In order to develop community-based strategies for earlier, targeted detection of liver disease, a good understanding is needed of which metabolic risk factors best predict severe NAFLD outcomes and advanced fibrosis. Research evidence published up until 2015 was synthesised to underpin the UK NICE guidelines [12], and this synthesis highlighted a paucity of evidence. Since then, several studies from large population cohorts have been published. In addition, the NICE review did not include all relevant outcomes. Cirrhosis and liver-related

mortality outcomes were omitted, though natural history studies suggest that it is reasonable to assume that people who develop liver cirrhosis or die from liver disease will have passed through the stage of non-alcoholic steatohepatitis (NASH) and advanced liver fibrosis. To address this gap in our understanding, we conducted an updated systematic review of published observational studies including all relevant outcomes. The aim was to synthesise evidence on which of the metabolic risk factors, or combination of risk factors, can best predict incident severe liver disease outcomes or NASH/advanced fibrosis in the general population at risk of NAFLD or with diagnosed NAFLD.

## Methods

### Registration of review protocol

The protocol for this review was registered in advance with PROSPERO (International Prospective Register of Systematic Reviews; CRD42018115459).

### Types of studies and inclusion and exclusion criteria

Original studies were included if they were observational, prospective, or retrospective studies that reported either (1) severe liver disease outcomes (cirrhosis, complications of cirrhosis, or liver-related death) or (2) NASH/advanced fibrosis in adults ($\geq$18 years old) with metabolic risk factors as compared with adult individuals without metabolic risk factors. Metabolic risk factors were defined as those included in the National Cholesterol Education Program Adult Treatment Panel III (NCEP ATP III) definition [15], with the addition of BMI $>$ 30 kg/m$^2$ as the most commonly measured obesity marker, assessed as individual risk factors or in combination, making up the metabolic syndrome.

We included both (1) studies where the cohort population had been predefined as having a diagnosis of NAFLD (based on ultrasound, coding, or abnormal liver blood tests in the absence of other diagnosed liver pathology) and (2) studies of general populations, if participants with risk factors for, or confirmed pathology from, alcohol, viral, or other liver disease were excluded or adjusted for.

The following types of studies were excluded: (1) studies where entry into the cohort was based on a tertiary referral and biopsy for clinical assessment of liver disease; (2) studies assessing only hepatocellular carcinoma as an outcome in the context of a non-cirrhotic liver; (3) studies using simple steatosis as an outcome; (4) studies performed in patients who had received liver transplants or were undergoing bariatric surgery; (5) studies where patients already had severe liver disease (as defined above) or NASH/advanced fibrosis at the time of cohort entry; and (6) studies that did not specifically report any odds ratio or hazard ratio (HR) with 95% CI for the outcome measure of interest.

We performed a systematic review following the Preferred Reporting Items for Systematic Reviews and Meta-Analyses (PRISMA) guidelines [16] (see attached checklist [S1 Table]).

### Search strategy and data extraction

Potentially relevant studies were identified through systematic literature searches of relevant databases (MEDLINE, EMBASE, the Cochrane Library, ClinicalTrials.gov, Conference Proceedings Citation Index–Science [CPCI-S; Web of Knowledge], and OpenGrey [http://www.opengrey.eu/]) in December 2018. No date or language restrictions were applied. Reference lists from potentially relevant papers and previous review articles were hand searched. MeSH (Medical Subject Headings) and free text terms for the metabolic risk factors and liver

outcomes of interest were used. The MEDLINE search strategy is available in S2 Table. Searches were updated in May 2019 and January 2020.

Two researchers (HJ and either GS or DS) independently screened titles and abstracts. Any disagreement in full-text selection was resolved by consensus. Record screening was also assisted by Rayyan, an online software tool that assesses similarities between selected records and highlights other potentially relevant studies based on the screener's previous selection [17]. Full texts of potentially relevant papers were obtained and read by 2 independent researchers with reference to the predefined set of criteria to determine final study inclusion. Data were extracted into a standardised, pre-piloted extraction form developed in Excel. For all studies, we extracted information on study design, source of data, prognostic factors of interest, outcomes of interest, and adjustment factors. Data extraction—undertaken by one researcher and checked by a second—was based on the updated Critical Appraisal and Data Extraction for Systematic Reviews of Prediction Modelling Studies checklist for prognostic studies (CHARMS-PF) [18].

### Assessment of risk of bias

Two authors (HJ and RB) assessed the risk of bias independently. Since the included studies were observational cohort studies of prognostic factors, the QUIPS (Quality in Prognosis Studies) tool was used [18]. The QUIPS tool allows for quality assessment in 6 domains: study participation, study attrition, prognostic factor measurement, outcome measurement, adjustment for other prognostic factors, and statistical analysis/reporting. Risk of bias rating is reported as low, moderate, or high for each domain and then an overall risk of bias assigned based on the ratings in each domain. Any discrepancies in rating were addressed by a joint re-evaluation with a third author.

### Data synthesis and analysis

The outcome measure for the meta-analysis was incident fatal and/or non-fatal severe liver disease in individuals with metabolic risk factors, in comparison with individuals without metabolic risk factors. The effect measures reported in the included studies were all HRs. The results of the studies were pooled, and an overall estimate of HR was obtained using a random-effects model. This model takes into account study heterogeneity, which was felt to be necessary from assessment of the clinical heterogeneity of the studies during data extraction, as well as the statistical heterogeneity as measured by the $I^2$ statistic. Where authors reported HRs for subgroups, a fixed-effects meta-analysis was first performed so a summary (pooled) HR could be included in the overall analysis. Publication bias was evaluated using visual inspection of funnel plots. Meta-analysis was carried out using Review Manager 5.3, Cochrane's meta-analysis software [19]. The prognostic factors with sufficient data and homogeneity between studies to carry out meta-analysis were T2DM and obesity (as measured by BMI). For each of these prognostic factors, severe liver disease outcomes were stratified into liver disease mortality, non-fatal severe liver disease events (cirrhosis and complications of cirrhosis), and a combined endpoint of both. Pre-specified sensitivity analyses were carried out to examine effect sizes when limiting the analysis to the following subgroups of studies: studies of participants with risk factors taken from a population with no previous diagnosis of NAFLD and studies with a low risk of bias as measured by the QUIPS tool. A narrative synthesis was conducted to expand on obesity as a prognostic factor of interest beyond BMI, and to summarise the evidence on the role of hypertension and lipid abnormalities in predicting advanced liver outcomes, as well as the evidence around combinations of metabolic risk factors for prognosticating advanced liver disease outcomes.

## Patient and public involvement

An expert patient and public involvement group, including patients with late diagnosed NAFLD, were involved in the design of this review. They have had no role in the conduct or reporting of this review but will be actively involved in dissemination of the results to regional and national patient support groups.

## Results

The searches identified 7,300 unique citations. Of the titles and abstracts screened, 267 articles were selected for full-text screening, where 245 were excluded for reasons reported in the PRISMA diagram (Fig 1). A total of 22 unique studies representing data from 16 cohorts were eligible for inclusion in the systematic review, and were assessed for quality [20–41]. Studies using data from the same cohort were only included if the sub-studies were assessing different prognostic metabolic factors.

## Characteristics of included studies

Studies were included from Europe (Sweden [21,22,26–28,33,37,38], UK [20,32], Italy [20], Netherlands [20], and Spain [20]), North America (US [23,25,29–31,34,39–41] and Canada [36]), and Asia (Singapore [24] and China [35]), with data on over 24 million individuals. All the eligible studies were prospective or retrospective cohorts in design, and were all community-based general population cohorts, some defined by data linkage. In 16 of the studies, representing 12 of the cohorts, the included population was not pre-selected on the basis of a diagnosis of NAFLD, and liver-related outcomes were presumed to represent outcomes from severe NAFLD, as participants with evidence of other common causes of liver disease were either excluded at cohort entry or adjusted for in the analysis. All of the included studies excluded individuals drinking alcohol at harmful levels and those with alcohol-related liver disease at cohort entry, or adjusted for alcohol consumption during analysis. In 6 of the studies, representing 4 of the cohorts, part of the population under study had a predefined diagnosis of NAFLD, defined using ultrasound, abnormal liver blood tests, or International Classification of Diseases (ICD) coding at the time of cohort entry [20,25,31,38–40]. In 8 of the studies, the cohort studied included only men or women, but was otherwise an approximately general population. Thirteen of the studies looked at T2DM as a prognostic factor of interest, 14 looked at BMI, and 4 were interested in other measures of central obesity. Fewer studies assessed the effects of dyslipidaemia and hypertension as individual metabolic risk factors, with heterogeneity in prognostic factor definition and outcome of interest. Metabolic syndrome as a risk factor was studied in 4 studies, with 3 of them based on sub-cohorts from the same population cohort [25,39,40]. Of the 22 studies, 5 employed liver disease mortality as an outcome measure, 11 fatal and non-fatal severe liver disease events (combined endpoint), and 6 non-fatal severe liver disease events (cirrhosis/complications of cirrhosis). None of the included studies reported NASH/advanced fibrosis as outcome measures although these were included in the search strategy. Liver disease events were validated in all the studies by medical records and death certificates using ICD diagnosis codes. Of the 22 included studies, 13 received a low risk of bias rating using the QUIPS tool (S3 Table). Further details of included studies are shown in Table 1.

## T2DM and the risk of incident severe liver disease events

Twelve studies were included in the meta-analysis of T2DM as a prognostic factor for incident severe liver disease. One study included in the review was excluded from the pooled primary

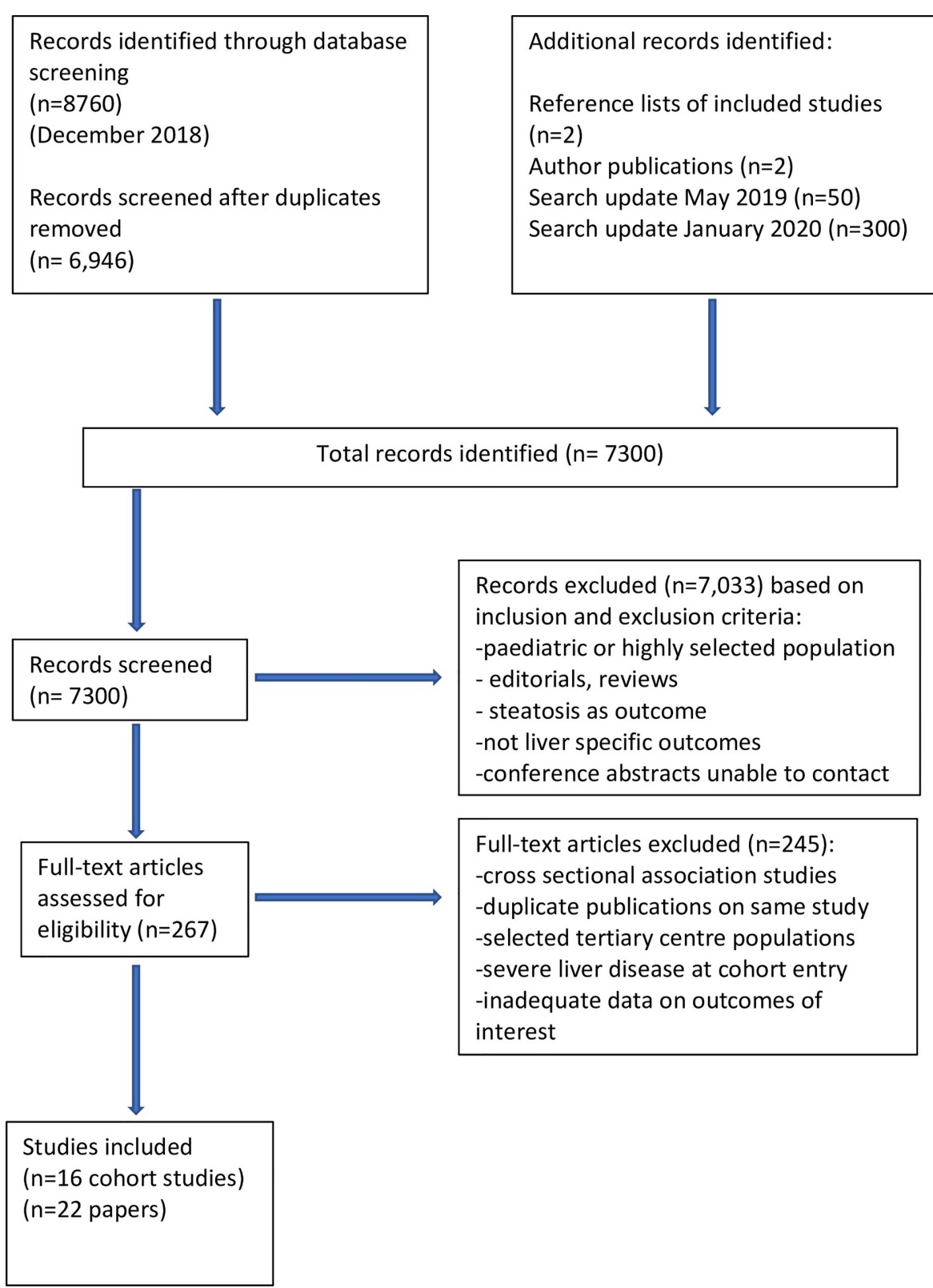

**Fig 1. PRISMA diagram of study selection.**

**Table 1. Characteristics of included studies.**

| Study | Country | Study design and population | Years of follow-up | Diagnosis of NAFLD at cohort inclusion | Metabolic RFs studied | Study outcomes of interest and number of events | Adjustments of interest considered | Adjusted HRs for liver events with 95% CIs and p-values | Risk of bias |
|---|---|---|---|---|---|---|---|---|---|
| Alexander 2019 [20] | UK, Netherlands, Italy, Spain | Retrospective data linkage cohort analysis, 18 million, 136,703 with NAFLD, mean age 55.8 years, 52% M | 3.3 | Yes (coding) | T2DM, BP, obesity | Cirrhosis/ complications, 7,375 events | Alcohol, other metabolic RFs | *BMI > 30: 1.03 (1.03–1.04), $p < 0.001$; T2DM: 2.86 (2.71–3.02), $p < 0.001$; high BP: 1.06 (1.00–1.12), $p = 0.03$ | Low |
| Andreasson 2017 [21] | Sweden (Malmo cohort) | Prospective population cohort, 27,617, mean age 58.1 years, 38.8% M | 19.8 | No (exclusion of other causes of LD) | Obesity | Composite non-fatal and fatal LD, 505 events | Alcohol | *BMI > 30: 1.52 (1.17–1.98), $p = 0.002$; increased WC: women: 1.75 (1.32–2.33), $p < 0.001$, men: 1.69 (1.28–2.23), $p < 0.001$; increased WHR: women: 1.68 (1.36–2.07), $p < 0.001$, men: 1.78 (1.41–2.25), $p < 0.001$ | Low |
| Björkström 2019 [22] | Sweden | Retrospective data linkage cohort analysis, 2.5 million, 406,770 with T2DM, mean age 64.7 years, 53.8% M | 7.7 | No (exclusion of other causes of LD) | T2DM | Composite non-fatal and fatal LD, 16,711 events | Unclear—high alcohol risk excluded at baseline | T2DM: 2.28 (2.21–2.36), $p < 0.001$ | Low |
| El-Serag 2004 [23] | US | Retrospective data linkage cohort analysis, 173,643 with diabetes, 650,620 without diabetes, age > 20 years, 98% M (veterans) | 10 | No (exclusion of other causes of LD) | T2DM | Composite non-fatal and fatal LD, 7,799 events | Alcohol | T2DM: 2.15 (2.00–2.31), $p < 0.001$ | Mod |
| Goh 2017 [24] | Singapore (Singapore Chinese Health Study) | Prospective population cohort, 63,247, age 45–74 years, 50% M | 16.9 | No (exclusion of other causes of LD) | T2DM, obesity | LD mortality, 133 events | Alcohol | T2DM: 2.6 (1.73–3.89), $p < 0.001$; BMI > 30: 1.36 (0.86–2.17), $p = 0.19$ | Low |
| Golabi 2018 [25] | US (NHANES III) | Prospective population cohort, 3,613, median age 43 years, 50% M | 19 | Yes (ultrasound) | Metabolic syndrome | LD mortality, 22 events | Alcohol | 1 MS RF: 26.35 (2.46–282.72), $p = 0.007$; 2 MS RF: 16.95 (1.59–180.91), $p = 0.019$; 3 MS RF: 1.98 (0.11–34.38), $p = 0.64$; 4 MS RF: 4.57 (0.32–64.88), $p = 0.26$ | Mod |
| Hagström 2016$ [26] | Sweden | Prospective population cohort, 44,248, age 18–20 years, 100% M (army conscripts) | 37.8 | No (exclusion of other causes of LD) | Obesity | Composite non-fatal and fatal LD, 393 events | Alcohol, BP | BMI > 30: 1.59 (0.64–3.95), $p = 0.32$ | Mod |
| Hagström 2018 [27] | Sweden | Prospective population cohort, 1,220,2161, age 17–19 years, 100% M (army conscripts) | 28.5 | No (exclusion of other causes of LD) | Obesity, T2DM | Composite non-fatal and fatal LD, 5,281 events | High alcohol risk excluded at baseline, obesity, BP | T2DM: 3.49 (3.01–4.03), $p < 0.001$ | Low |

(*Continued*)

**Table 1.** (*Continued*)

| Study | Country | Study design and population | Years of follow-up | Diagnosis of NAFLD at cohort inclusion | Metabolic RFs studied | Study outcomes of interest and number of events | Adjustments of interest considered | Adjusted HRs for liver events with 95% CIs and *p*-values | Risk of bias |
|---|---|---|---|---|---|---|---|---|---|
| Hagström 2019 [28] | Sweden | Retrospective data linkage cohort analysis, 1,185,733, mean age 28.6 years, 100% F (antenatal) | 13.8 | No (sensitivity analysis to exclude alcohol diagnoses) | Obesity, T2DM | Composite non-fatal and fatal LD, 852 events | Obesity, T2DM | BMI > 30: 1.76 (1.27–2.46), *p* = 0.001; T2DM: 4.30 (3.23–5.72), *p* < 0.001 | Low |
| Ioannou 2003 [29] | US (NHANES I) | Prospective population cohort, 11,465, age 25–74 years, 50% M | 13 | No (exclusion of cirrhosis from other causes of LD) | Obesity | Composite non-fatal and fatal LD, 89 events | T2DM, cholesterol, alcohol | BMI > 30: 1.65 (0.9–3.1), *p* = 0.11 | Low |
| Ioannou 2005 [30] | US (NHANES I) | Prospective population cohort, 11,434, age 25–74 years, 50% M | 13 | No (exclusion of cirrhosis from other causes of LD) | Obesity (central) | Composite non-fatal and fatal LD, 88 events | Alcohol | BMI > 30 and subscapular-to-triceps skinfold thickness ratio: high: 2.2 (1.1–4.6), *p* = 0.026, low: 0.8 (0.2–2.8), *p* = 0.75 | Low |
| Kanwal 2019 [31] | US | Retrospective data linkage cohort analysis, 271,906, mean age 55.5 years, 94.3% M | 9 | Yes (abnormal blood tests) (exclusion of other causes of LD) | All metabolic risk factors | Cirrhosis, 22,794 events | Alcohol risk excluded at baseline and throughout follow-up period, other metabolic RFs | BMI > 30: 1.09 (1.06–1.13), *p* < 0.001; T2DM: 1.31 (1.27–1.34), *p* < 0.001; high BP: 1.59 (1.51–1.69), *p* < 0.001; dyslipidaemia (composite): 1.23 (1.19–1.28), *p* < 0.001; 2 MS RF: 1.33 (1.26–1.40), *p* < 0.001; 3 MS RF: 1.61 (1.53–1.69), *p* < 0.001; 4 MS RF: 2.03 (1.93–2.13), *p* < 0.001 | Low |
| Liu 2010 [32] | UK (Million Women Study) | Prospective population cohort, 1,230,662, mean age 56 years, 100% F | 6.2 | No (exclusion of other causes of LD) | Obesity | Composite non-fatal and fatal LD, 1,811 events | Alcohol, BMI, T2DM | BMI > 30: 1.49 (1.33–1.68), *p* < 0.001; T2DM: 4.29 (2.74–6.73), *p* < 0.001 | Low |
| Nderitu 2017 [33] | Sweden (AMORIS cohort) | Prospective population cohort, 509,436, mean age 44 years, 53.4% M | 20 | No (exclusion of other causes of LD) | All metabolic risk factors | Cirrhosis/ complications, 2,775 events | Other metabolic RFs | low HDL: 1.28 (1.04–1.59), *p* = 0.020; high triglycerides: 1.30 (0.99–1.72), *p* = 0.059; BMI > 30: 1.38 (0.93–2.04), *p* = 0.11; T2DM: 2.00 (1.19–3.38), *p* = 0.009 | Mod |
| Otgonsuren 2013 [34] | US (NHANES III) | Prospective population cohort, 10,565, age 20–50 years, 45% M | 13.8 | Yes (ultrasound) (exclusion of other causes of LD) | Obesity | LD mortality, 26 events | Alcohol, BP, T2DM | BMI > 30: 1.06 (0.96–1.16), *p* = 0.25; WC: 1.02 (0.98–1.07), *p* = 0.332; WHR > 0.8: 83.51 (2.03–3,434.26), *p* = 0.02 | Low |
| Pang 2018 [35] | China (China Kadoorie Biobank) | Prospective population cohort, 503,993, mean age 51.5 years, 41% M | 10 | No (exclusion of other causes of LD) | T2DM | Cirrhosis/ complications, 2,082 events | Alcohol, BMI | T2DM: 1.78 (1.45–2.18), *p* < 0.001 | Mod |

(*Continued*)

**Table 1.** (Continued)

| Study | Country | Study design and population | Years of follow-up | Diagnosis of NAFLD at cohort inclusion | Metabolic RFs studied | Study outcomes of interest and number of events | Adjustments of interest considered | Adjusted HRs for liver events with 95% CIs and p-values | Risk of bias |
|---|---|---|---|---|---|---|---|---|---|
| Porepa 2010 [36] | Canada | Retrospective data linkage cohort analysis, 2,497,777, mean age 55.3 years, 56.3% M | 6.4 | No (exclusion of other causes of LD) | T2DM, BP, obesity | Cirrhosis/ complications, 8,365 events | BP, lipids, obesity, T2DM | T2DM: 1.77 (1.68–1.86), $p < 0.001$; high BP: 1.23 (1.14–1.31), $p < 0.001$; BMI > 30: 1.16 (1.01–1.33), $p = 0.03$ | Low |
| Schult 2011 [37] | Sweden (Gothenberg survey) | Prospective population cohort, 855, mean age 50 years, 100% M | 40 | No (exclusion of other causes of LD) | All metabolic risk factors | Composite non-fatal and fatal LD, 14 events | Alcohol | BMI > 30: 1.27 (1.09–1.48), $p = 0.002$; triglycerides: 1.99 (1.35–2.96), $p = 0.001$; other HRs not presented | Mod |
| Schult 2018 [38] | Sweden (Gothenberg survey) | Prospective population cohort, 1,462, age 38–60 years, 100% F | 42 | No (exclusion of other causes of LD) | Obesity (central) | Composite non-fatal and fatal LD, 11 events | Alcohol, BP | WHR > 0.8: 5.82 (1.59–21.4), $p = 0.008$ | Mod |
| Simeone 2017 [39] | US | Retrospective data linkage cohort analysis, 18,754, age > 18 years, 38.5% M | 2.3 | Yes (coding) (exclusion of other causes of LD) | T2DM | Composite non-fatal and fatal LD, 5,645 events (any disease progression) | Unclear | T2DM: 2.0 (no CI given) | High |
| Stepanova 2010 [40] | US (NHANES III) | Prospective population cohort, 991, age > 17 years, 47.5% M | 13.3 | Yes (abnormal blood tests) (exclusion of other causes of LD) | All metabolic risk factors | LD mortality, 117 events | Alcohol, other metabolic RFs | T2DM: 1.05 (1–1.65), $p < 0.05^{@}$; high cholesterol: 0.37 (0.06–2.15), $p = 0.284$; high BP: 0.07 (0.01–0.3x), $p = 0.007$; BMI > 30: 11.19 (2.43–51.56), $p = 0.002$; MS: 12.08 (1.10–132.22), $p = 0.042$ | Mod |
| Younossi 2013 [41] | US (NHANES III) | Prospective population cohort, 1,448, age > 18 years, 64% M | 16 | Yes (ultrasound) (exclusion of other causes of LD) | Metabolic syndrome, obesity | LD mortality, 10 events | Metabolic RFs, alcohol | BMI > 30: 1.12 (1.03–1.21), $p = 0.008$; MS: 294.24 (118.74–729.14), $p < 0.001$ | Low |

BMI units are kg/m$^2$.

*Adjusted HR for whole cohort using a fixed-effects meta-analysis to get the combined HR from the 2 presented HRs for subgroups (coded versus uncoded combined in Alexander et al.; men and women combined in Andreasson et al.).

$Sub-cohort of the 2018 Hagström study but with additional data on alcohol consumption.

@CI and p-value as given in the paper presented here—different from the calculated CI used in meta-analysis using the HR and sample size (Fig 2). This difference is due to CI asymmetry in the published figures and inability to reproduce these figures on log transformation. Authors contacted to confirm data—no response.

BMI, body mass index; BP, blood pressure; CI, confidence interval; F, female; HDL, high-density lipoprotein; HR, hazard ratio; LD, liver disease; M, male; mod, moderate; MS, metabolic syndrome; NAFLD, non-alcoholic fatty liver disease; RF, risk factor; T2DM, type 2 diabetes mellitus; WHR, waist-to-hip ratio; WC, waist circumference.

analysis as insufficient data were presented to support calculation of confidence intervals around the adjusted effect measure [39]. Overall, in the 12 observational studies, there were 22.8 million individuals followed up for a median of 10 years (IQR 6.4–16.9) experiencing 72,792 fatal and/or non-fatal severe liver disease events. Most of the studies included middle-

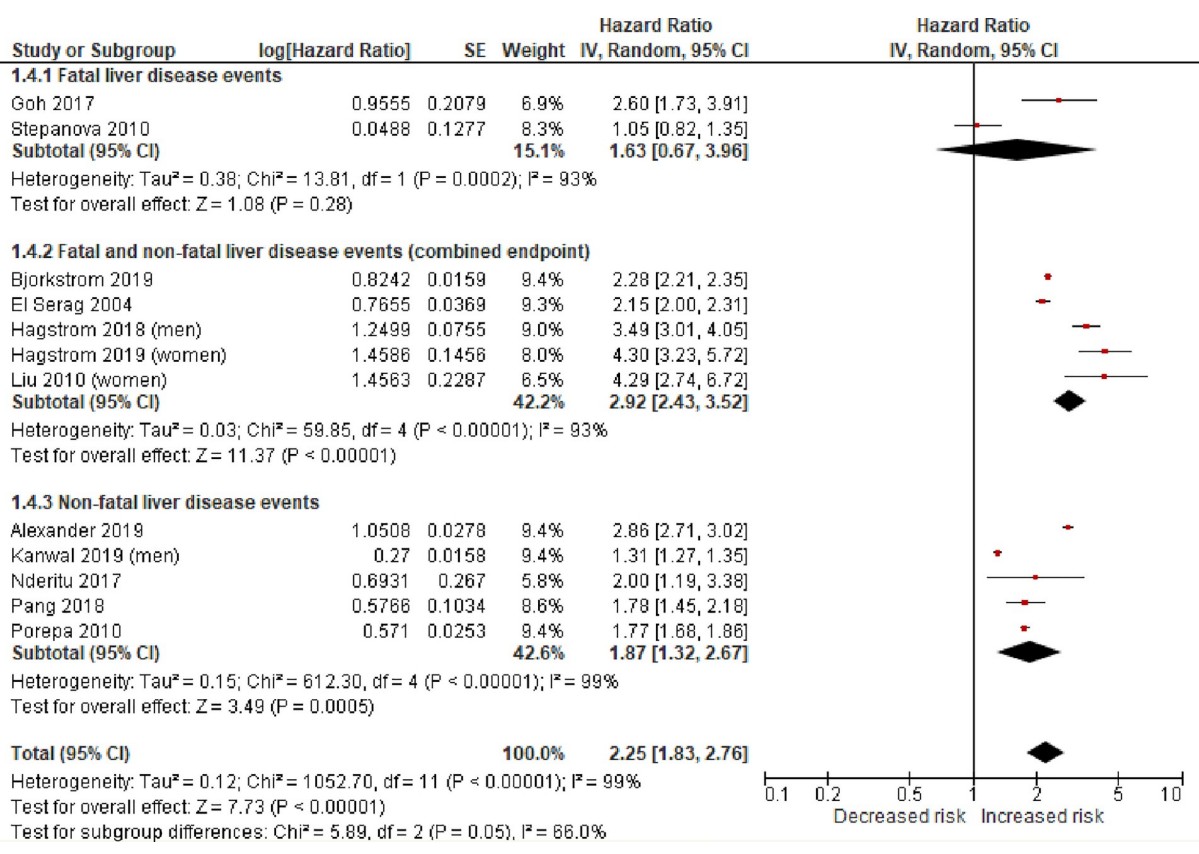

**Fig 2. Random-effects meta-analysis of the risk of incident severe liver disease associated with type 2 diabetes.** Statistical test for study heterogeneity = chi-squared test. Statistical test for summary effect in the meta-analysis = $Z$ test. IV, inverse variance.

aged individuals, with 7 studies including male and female individuals in roughly equal numbers, 2 studies including only women, and 3 only/predominantly men.

The individual study and pooled estimates of the association between T2DM and risk of severe liver disease are shown in Fig 2. T2DM was significantly associated with an increased risk of severe liver disease events (random-effects HR 2.25, 95% CI 1.83–2.76, $p < 0.001$, $I^2$ 99%). There was no asymmetry of the funnel plot to suggest a publication bias (S1 Fig).

## Obesity and the risk of incident severe liver disease events

Fourteen studies were included in the meta-analysis of obesity as a prognostic factor for incident severe liver disease. The definition of obesity used for the meta-analysis was a BMI > 30 kg/m$^2$ as this was the most widely reported metric used. Some of the included studies, and others, also looked at alternative measures of obesity risk, such as waist-to-hip ratio (WHR). There were too few studies to pool these results, but the findings are reported in the narrative synthesis below. The 14 observational studies in the meta-analysis provided data on 19.3 million individuals followed up for a median of 13.8 years (IQR 9.0 to 19.8) experiencing 49,541 fatal and/or non-fatal severe liver disease events. Nine of the studies of predominantly middle-aged individuals included men and women in roughly equal numbers. Two studies looked at women only, with 1 of the cohorts recruiting women in the early stages of pregnancy only [28]. The 3 remaining studies recruited predominantly men—1 at army conscription (ages 18–20 years), producing a younger study population at baseline, with follow-up for nearly 40 years [26].

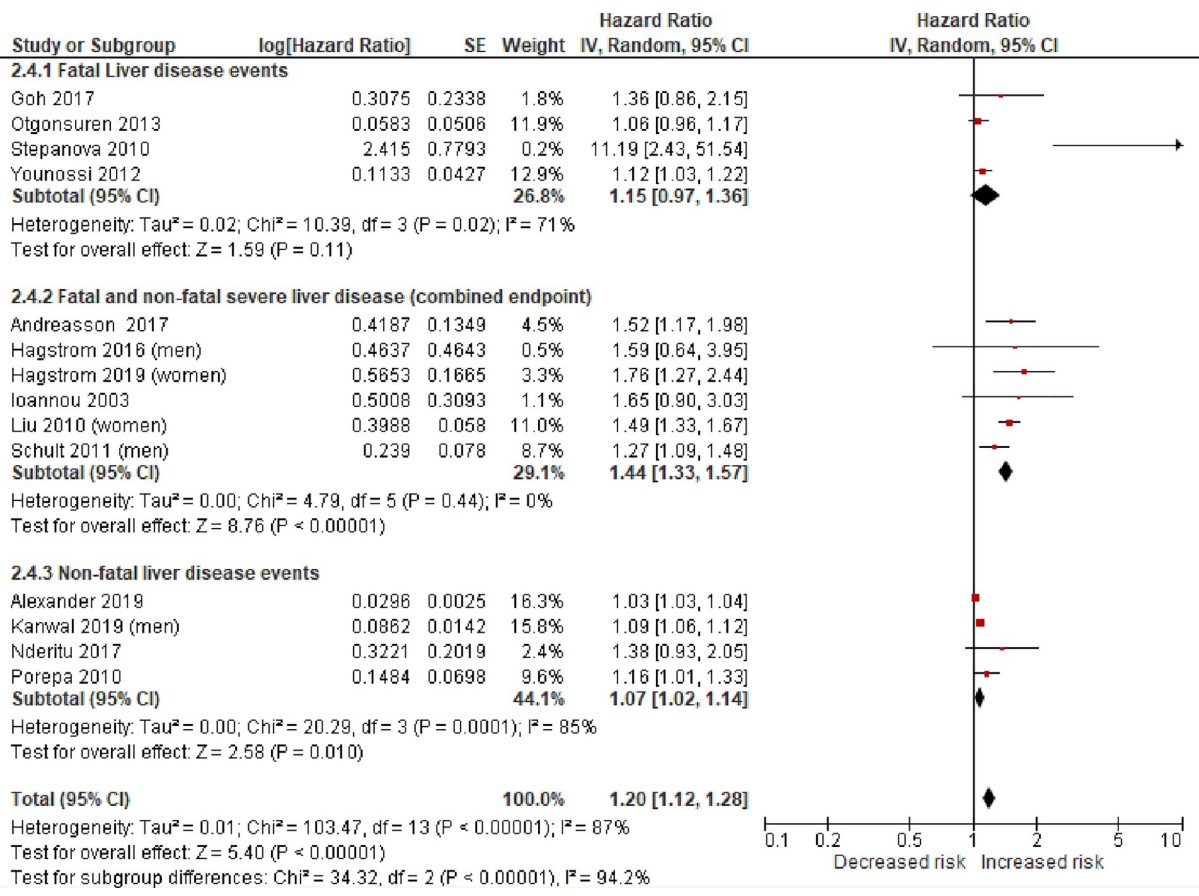

**Fig 3. Random-effects meta-analysis of the risk of incident severe liver disease associated with obesity (BMI > 30 kg/m$^2$).** Statistical test for study heterogeneity = chi-squared test. Statistical test for summary effect in the meta-analysis = $Z$ test. IV, inverse variance.

The individual and pooled estimates of association between obesity (BMI > 30 kg/m$^2$) and risk of severe liver disease are shown in Fig 3. A BMI > 30 kg/m$^2$ was associated with an increased risk of severe liver disease events (random-effects HR 1.20, 95% CI 1.12–1.28, $p <$ 0.001, $I^2$ 87%). There was some asymmetry of the funnel plot, suggesting possible under-publishing of smaller negative studies (S2 Fig).

## Sensitivity analyses

Limiting the analysis to studies judged to be at low risk of bias and excluding studies where NAFLD was diagnosed at cohort entry provided overall estimates consistent with the primary analysis for both prognostic factors that were meta-analysed (Table 2). The high levels of heterogeneity, as indicated by the high $I^2$ values, were explored. These were felt to be due to the variation in study design, particularly around the range of populations and outcomes studied, leading to clinical heterogeneity. Despite this, there was a consistent direction of effect, and, based on the objective of the review, pooling using meta-analysis was still felt to be appropriate.

## Other measures of central obesity

Four studies looked at alternative measures of central obesity as possible prognostic factors for severe liver disease outcomes. One prospective cohort examined the subscapular-to-triceps

**Table 2.  Risk of fatal and/or non-fatal severe liver disease events associated with T2DM and obesity: Sensitivity analyses.**

| Analysis | Number of comparisons | Overall adjusted HR with 95% CI | $I^2$ value |
|---|---|---|---|
| **T2DM and risk of severe liver disease** | | | |
| Including only those with no previous diagnosis of NAFLD at cohort entry | 10 | 2.54 (2.19–2.94), $p < 0.001$ | 96% |
| Including only studies with low risk of bias using QUIPS tool | 8 | 2.59 (1.99–3.36), $p < 0.001$ | 99% |
| **BMI > 30 kg/m$^2$ and risk of severe liver disease** | | | |
| Including only those with no previous diagnosis of NAFLD at cohort entry | 11 | 1.29 (1.14–1.46), $p < 0.001$ | 87% |
| Including only studies with low risk of bias using QUIPS tool | 10 | 1.18 (1.10–1.26), $p < 0.001$ | 89% |

CI, confidence interval; HR, hazard ratio; NAFLD, non-alcoholic fatty liver disease; QUIPS, Quality in Prognosis Studies; T2DM, type 2 diabetes mellitus.

skinfold thickness ratio (SFR) as a measure of central obesity [30], concluding that in obese individuals (BMI > 30 kg/m$^2$), only those with a SFR > 1 were at increased risk of a combined fatal/non-fatal severe liver disease outcome (HR 2.2, 95% CI 1.1–4.6, $p = 0.026$). Two studies reported the association between waist circumference (WC) and liver disease events [21,34], with 1 of the studies (using a combined fatal/non-fatal endpoint) reporting that a WC over 88 cm in women was a better predictor of liver outcomes than BMI (HR for BMI > 30 kg/m$^2$: 1.3, 95% CI 0.4–1.88, $p = 0.16$; HR for WC > 88: 1.75, 95% CI 1.32–2.33, $p < 0.001$), but that this was not the case for men [21]. The other study found no significant association between WC and liver disease deaths, but did not stratify results by sex [34]. Two studies analysed the relationship between WHR and severe liver disease outcomes. One study focused on women and, using a combined fatal/non-fatal endpoint, found a strong association between a WHR > 0.8 and severe liver disease (HR 5.82, 95% CI 1.59–21.4, $p = 0.008$). Only a small number of the nearly 1,500 cohort participants had diabetes recorded at cohort entry ($n = 13$), and no incident diabetes was recorded during follow-up. The lack of meaningful adjustment for diabetes was felt to be a study weakness [38]. The other study reporting WHR as a prognostic factor again found this central obesity measure to prognosticate better than BMI in women only, with the HR being nearly identical to that for BMI > 30 kg/m$^2$ in men. For women, the HR was 2.05 (95% CI 1.49–2.82, $p < 0.001$) for those with a WHR more than 0.05 above normal [21]. There were insufficient similar studies to be able to pool any of the results, but the available data suggest that measures of central obesity are better at prognosticating for severe liver disease outcomes than BMI alone, particularly in women.

## Other metabolic risk factors and the risk of severe liver disease events

**Lipids.**  Five studies investigating lipid levels, and their prognostic value for liver disease outcomes, looked at low high-density lipoprotein (HDL), high triglycerides, combined lipid abnormalities, and hypercholesterolaemia as exposures of interest, with varying cutoff points for 'abnormality', so direct comparison and pooling was not attempted. By far the largest study that looked at low HDL and high triglycerides as independent risk factors, in line with cutoffs for a diagnosis of metabolic syndrome, examined an unselected population of over 100,000. This study reported a HR for a non-fatal severe liver disease event of 1.28 for low HDL (95% CI 1.04–1.59, $p = 0.02$) and 1.30 for high triglycerides (95% CI 0.99–1.72, $p = 0.059^*$; analysis done on a smaller dataset of 65,000 with available complete data) [33]. A large population-based data linkage study of over 270,000 individuals (over 95% male) supports these findings, with a reported HR of 1.23 (95% CI 1.19–1.28, $p < 0.001$) for an outcome of cirrhosis using a combined dyslipidaemia exposure based on low HDL and/or high triglycerides [31]. This suggests a smaller adjusted effect of these metabolic risk factors compared to the effect of T2DM, perhaps similar to the adjusted effect of a BMI > 30 kg/m$^2$, but is based on few studies.

**Hypertension.** Four studies reported on hypertension as a prognostic factor of interest in predicting severe liver outcomes. A mortality study looking at individuals with presumed NAFLD (based on abnormal liver blood tests) found a negative association after adjustment for other metabolic risk factors (HR 0.07, 95% CI 0.01–0.3, $p = 0.007$) [40]. This is contradicted by 2 larger population-based data linkage studies looking at non-fatal severe liver disease, which both report a positive association between diagnosed hypertension and an incident liver outcome with HRs of 1.23 (95% CI 1.14–1.31, $p < 0.001$) [36] and 1.59 (95% CI 1.51–1.69, $p < 0.001$) [31]. This association was supported, although with a much smaller effect size, by findings of a study using several large European primary care datasets to report non-fatal liver outcomes (HR 1.06, 95% CI 1.00–1.12, $p = 0.03$) [20].

**Metabolic syndrome.** Three articles reported on the association between metabolic syndrome (NCEP ATP III definition) [15] and liver mortality using data from the same population cohort study [25,40,41], with 1 additional study looking at combined metabolic risk with cirrhosis as the outcome [31]. The effect sizes in the mortality data are inconsistent, with very wide confidence intervals, despite the studies representing the same population. One of the studies reported a weakening of the association of metabolic risk factors with liver-related mortality with increasing number of metabolic risk factors [25]. The other 2 analyses concluded that metabolic syndrome (≥3 metabolic risk factors) was associated with an increased risk of liver-related mortality, with reported HRs of 12.08 (95% CI 1.10–132.22, $p = 0.042$) [40] and 294.24 (95% CI 118.74–729.14, $p < 0.001$) [41]. A more recent, larger population data linkage study looked in detail at combinations and numbers of metabolic risk factors associated with cirrhosis outcomes, reporting increasing HRs for increasing numbers of risk factors, with a HR of 2.56 (95% CI 2.26–2.92, $p < 0.001$) for those with T2DM, obesity, hypertension, and dyslipidaemia [31].

## Discussion

In this systematic review and meta-analysis of 22 studies including data from over 24 million individuals, we found that T2DM was significantly associated with incident severe liver disease, with a more than 2-fold increase in the combined outcomes studied (random-effects HR 2.25, 95% CI 1.83–2.76, $p < 0.001$, $I^2$ 99%). There was a less marked association between obesity and incident severe liver disease using BMI > 30 kg/m$^2$ as the obesity measure (random-effects HR 1.20, 95% CI 1.12–1.28, $p < 0.001$, $I^2$ 87%), with a suggestion that other measures of central adiposity may better predict poor liver outcomes, particularly in women.

There were many fewer studies looking at the relationship between other metabolic risk factors and incident severe liver disease, with differing definitions of prognostic factors of interest. Pooling of results was therefore not appropriate, but the suggestion from the largest, highest quality studies was that lipid abnormalities (low HDL and high triglycerides) and hypertension are both independently associated with incident severe liver disease. The adjusted effect sizes appear to be similar to that for high BMI. Fewer data were available looking at combinations of metabolic risk factors making up the metabolic syndrome as a predictor of liver outcomes, with a suggestion from the largest study of an increase in non-fatal liver outcomes in those with metabolic syndrome of a similar magnitude to that for T2DM.

The presented review focuses on general population data, aiming for the results to be applicable for clinicians seeing unselected patients. Studies of individuals with biopsy-proven NAFLD at cohort entry have been criticised due to the inherent bias of selecting patients who have been referred for liver biopsy, and the relatively short median follow-up time. These studies, however, provide important comparative and supportive evidence.

Studies looking at metabolic risk factors in patients with biopsy-proven NAFLD and long-term severe liver disease outcomes have found strong independent associations between

T2DM at the time of biopsy and liver-related outcomes [4,41–43]. HRs for T2DM as a predictor of severe outcomes in these studies vary more widely, partly due to the inclusion of all-cause mortality in some of the studies, where the commonest cause of death was cardiovascular disease rather than liver-related mortality. Studies specifically reporting liver-related outcomes clearly report T2DM as the most important clinical risk factor, reporting HRs between 2.19 (95% CI 1.00–4.81) [43] and 22.83 (95% CI 2.97–175.03) [44] for liver-related mortality. These studies generally do not report other metabolic risk factors as independent predictors of poor outcome, although smaller sample sizes may indicate they were not powered adequately to detect these smaller risk increases.

A significant body of related research has come from paired sequential liver biopsy studies looking at the association between metabolic risk factors and histological NAFLD progression. A systematic review of 11 paired biopsy studies (411 individuals) published in 2015 indicated that only hypertension was significant in predicting the rate of histological progression between biopsies (odds ratio 1.94, 95% CI 1.00–3.74) [45]. In line with our findings, more recent studies, including the largest single-centre biopsy cohort to date, identified T2DM as the strongest metabolic predictor of histological disease progression [46,47].

A large body of work has been extensively reviewed and synthesised on the epidemiology and natural history of NAFLD. The focus of these reviews is distinct yet complementary to our work. They identify the high and rising global burden of NAFLD and associated adverse outcomes using prevalence data from cross-sectional studies of people with a confirmed diagnosis of NAFLD. These reviews estimate the global prevalence of NAFLD in people with diabetes to be more than double that of the general population (55.48% versus 25.2%) [1,48,49]. A recent meta-analysis looking at NAFLD in T2DM reported prevalence estimates for NASH of 37.3% and advanced fibrosis of 17% in those with T2DM, far higher than general population estimates of these progressive forms of NAFLD [49]. This review adds to these prevalence data, indicating that the rate of incident severe liver outcomes is also significantly higher in those with T2DM.

It is noteworthy that despite our outcome inclusion criteria including NASH and advanced fibrosis, none of the included studies reported these earlier disease stages as outcomes. This leaves information on the association between metabolic risk and NASH/advanced fibrosis coming from cross-sectional and highly selected populations [48]. As NASH and advanced fibrosis have traditionally been histological diagnoses requiring a liver biopsy, this is not surprising and may explain the paucity of evidence reported in a similar review of the ability of NAFLD risk factors to predict progressive disease in the population [12].

In this synthesis we included data from population cohorts without a definite clinical diagnosis of NAFLD at baseline. It is therefore possible that not all liver outcomes in these groups were due to underlying NAFLD, which is a study limitation. All included studies reported that people with known liver disease of other common aetiologies (which would include viral hepatitis) were excluded and have adjusted for alcohol in the analysis. However, the possibility of other undiagnosed pathologies cannot be fully excluded. A recent multi-site European cohort study found that metabolic risk factors predicted cirrhosis with similar effect sizes for people with and without a coded diagnosis of NAFLD [20] and suggested this was likely due to NAFLD not being diagnosed or accurately coded (i.e., hidden disease in the control group). Other studies have also reported lower than expected levels of diagnostic coding for NAFLD [50]. This suggests that our approach may be a strength, as only a minority of people living with NAFLD have had a formal diagnosis, and so represent a highly selected subgroup.

The limitations of synthesising observational data, including the issue of unmeasured confounding, are well known, and the clinical and statistical heterogeneity described in this review

was not unexpected. We also acknowledge the possibility of publication bias. However, this was a large study, including data on over 24 million individuals with over 300 million person years of follow-up. Use of predetermined inclusion and exclusion criteria and robust quality assessment mean we have included the best available evidence to report on the outcome of incident advanced liver disease related to metabolic risk.

Identifying those at risk of severe liver disease in the community setting will only be beneficial if effective lifestyle interventions and/or liver-targeted medications are effective and available. We have increasing evidence for the clinical effectiveness of lifestyle interventions in NAFLD [51,52], 1 compound already has demonstrated efficacy in a phase III trial [53], and several other promising liver-targeted medications are also in phase III studies [54]. There is also an increasing evidence base around the cost-effectiveness of earlier case finding in the community setting [11,55]. These advances highlight the timely nature of this review, which can help guide clinicians and primary care policy-makers towards selecting the patients most likely to benefit from these interventions. Future research should focus on studying prospective population cohorts for earlier liver outcomes and their relationship to metabolic risk, including the interplay of these risk factors in combination. With increased availability of non-invasive methods to look for advanced fibrosis, looking at earlier outcomes will become both more realistic for research studies and, more importantly, more relevant for clinicians managing unselected populations who are looking to target, diagnose, and manage those at increased risk of poor outcomes before they develop decompensated cirrhosis.

In conclusion, this robust meta-analysis provides evidence to suggest that people with T2DM have a significantly increased risk of future severe liver disease and that obesity (as measured by BMI) also has an impact on risk. More evidence is needed around the interplay of metabolic risk factors (metabolic syndrome) in predicting severe liver outcomes in people at risk of NAFLD. Our findings support a more structured, risk-factor-based approach in NAFLD management, particularly for patients with T2DM.

## Supporting information

**S1 Fig. Funnel plot of studies included in the T2DM meta-analysis.**
(TIF)

**S2 Fig. Funnel plot of studies included in the obesity meta-analysis.**
(TIF)

**S1 Table. PRISMA checklist.**
(DOCX)

**S2 Table. MEDLINE search strategy.**
(DOCX)

**S3 Table. Risk of bias of included studies.**
(DOCX)

## Acknowledgments

Transparency declaration: I (HJ) confirm that this manuscript is an honest, accurate, and transparent account of the study being reported; that no important aspects of the study have been omitted; and that any discrepancies from the study as planned have been explained. No ethical approval was required for this systematic review.

## Author Contributions

**Conceptualization:** Helen Jarvis, Quentin M. Anstee, Barbara Hanratty.

**Data curation:** Helen Jarvis, Robert Barker, Gemma Spiers, Daniel Stow.

**Formal analysis:** Helen Jarvis, Dawn Craig.

**Investigation:** Helen Jarvis, Robert Barker, Gemma Spiers, Daniel Stow.

**Methodology:** Dawn Craig.

**Supervision:** Dawn Craig, Quentin M. Anstee, Barbara Hanratty.

**Writing – original draft:** Helen Jarvis.

**Writing – review & editing:** Dawn Craig, Robert Barker, Gemma Spiers, Daniel Stow, Quentin M. Anstee, Barbara Hanratty.

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
