## [Decision Letter · Decision Letter 0]

24 Dec 2019

Dear Dr. Jarvis,

Thank you very much for submitting your manuscript "Metabolic risk factors and incident advanced liver disease in NAFLD: A systematic review and meta-analysis of population based observational studies" (PMEDICINE-D-19-04140) for consideration at PLOS Medicine. 

[LINK]

In light of these reviews, I am afraid that we will not be able to accept the manuscript for publication in the journal in its current form, but we would like to consider a revised version that addresses the reviewers' and editors' comments. Obviously we cannot make any decision about publication until we have seen the revised manuscript and your response, and we plan to seek re-review by one or more of the reviewers. 

We expect to receive your revised manuscript by Jan 14 2020 11:59PM. Please email us (plosmedicine@plos.org) if you have any questions or concerns.

We look forward to receiving your revised manuscript. 

Sincerely,

Louise Gaynor-Brook, MBBS PhD

Associate Editor 

PLOS Medicine

plosmedicine.org

Please provide p vaues in the abstract (and main text and tables) as well as removing funding information from the abstract.

Abstract and throughout – please remove causal language such as “This review demonstrates that T2DM leads to a greater than two fold increase in the risk of developing severe liver disease. “ Causality cannot be shown from systematic reviews.

“Structured abstract” should simply read “Abstract”

Line 174 “Two researchers (HJ, GS or DS)” surely 3 researchers? Also line 183

Line 354 – remove bold font, please

Please present references in square brackets (rather than superscript) at the end of each sentence, prior to the full stop.

Abstract

Please report your abstract according to PRISMA for abstracts, following the PLOS Medicine abstract structure (Background, Methods and Findings, Conclusions) http://www.plosmedicine.org/article/info:doi/10.1371/journal.pmed.1001419

Abstract Background: Provide expand on the context of why the study is important. 

Abstract Methods and Findings: Please provide the dates of search, synthesis/appraisal methods and main outcome measure(s). In the last sentence of the Abstract Methods and Findings section, please describe the main limitation(s) of the study's methodology. In the last sentence of the Abstract Methods and Findings section, please describe the main limitation(s) of the study's methodology.

Please begin your Abstract Conclusions with “In this study, we observed ..." or similar. Please avoid vague statements such as "this finding requires a health policy response", mentioning only specific implications substantiated by the results.

Please remove subheading ‘Main text’

Introduction: If there has been a systematic review of the evidence related to your study (or you have conducted one), please refer to and reference that review and indicate whether it supports the need for your study. 

Methods

When completing the PRISMA checklist, please use section and paragraph numbers, rather than page numbers.

Please update your search to the present time. We require that SRs are updated to within roughly 6 months of the expected publication date. 

Results

Line 144 - please define NCEP ATP 

Line 182 - please define CHARMS-PF

Line 402 - please revise to ‘than’

Figures 2 & 3 - When a p value is given, please specify the statistical test used to determine it (in the figure legend) 

Table 1 & 2 - please define all abbreviations in the table legend. 

Discussion

Please present and organize the Discussion as follows: a short, clear summary of the article's findings; what the study adds to existing research and where and why the results may differ from previous research; strengths and limitations of the study; implications and next steps for research, clinical practice, and/or public policy; one-paragraph conclusion.

References

Please provide the names of the first 6 authors prior to ‘et al’ being used

Please ensure that journal titles are consistent e.g. Lancet/The Lancet 

Comments from the reviewers:

Reviewer #1: In this systematic review and meta-analysis, the authors analyze metabolic risk factors and their potential to predict liver disease outcomes in those with NAFLD and in the general population at risk for NAFLD. The authors find that type 2 diabetes leads to a greater than 2 fold increase in the risk of developing severe liver disease. It is well written, includes an extensive search for relevant studies, and with a large number of patients. 

My comments are the following:

- Although AASLD and NICE to don't recommend screening for NAFLD, would add that the American Diabetes Association most recent guidelines do recommend screening for fatty liver disease and advanced fibrosis in those with diabetes. This complements the findings of this study nicely.

- I am concerned about the inclusion criteria - there are only a few studies that have a specific diagnosis of NAFLD; the rest include only general populations at risk of NAFLD after excluding those with alcohol history and other causes of liver disease as the authors state in the methods and table 1. However, I reviewed one of the articles included in the meta-analysis (Andreasson 2017) and it does not appear that viral hepatitis or other types of liver disease were excluded. Can the authors comment on this? In addition, there are some studies that do not explicitly state that other liver disease was excluded (i.e. Liu 2010) - why was this? This is important as underlying liver disease (i.e. hepatitis c) will have a compounding effect on those with diabetes to result in progressive / advanced liver disease. By not excluding other causes of liver disease, this no longer becomes a study about NAFLD but rather a general population study.

- Also in table 1, the column with "? Diagnosis of NAFLD at baseline" is unclear - what is the question?

- In table 1 - please include the reference for each study after author/year.

- While I understand the authors' point that only a minority of people living with NAFLD have had a formal diagnosis and that these patients maybe a highly selected group, I still find the title of the study misleading - instead of the NAFLD population, I would argue that this is a general population; yes, the majority of these patients likely have NAFLD, but without a linking diagnosis (through coding, liver tests, imaging), one cannot make that conclusion.

Overall an excellent study with very strong public health implications.

Reviewer #2: Jarvis et al have performed a systematic review and meta-analysis on risk factors for development of severe liver disease, defined as fatal or non-fatal ICD-coding corresponding to cirrhosis or complications thereof. Main results are that primarily T2D is the major risk factor for severe liver disease, with obesity also a significant risk factor. 

The review and analyses is appropriate, and gives somewhat more precise estimates than the individual studies. However, I think the paper could be strengthened, if possible, by adding sex-specific analyses (is the effect of T2D/obesity similar in men and women?) and by age categories? Also, if possible would be interesting to see effect of T2D + obesity. Could be added to Table 2. 

Minor: Ref #46 seem to be an abstract from a conference. Please refer to the final paper. 

Reviewer #3: Alex McConnachie, Statistical Review

The paper by Jarvis et al presents the results of a systematic review and meta-analysis of incident advanced liver disease outcomes and metabolic risk factors. This review considers the use of statistics in the paper. Overall, these are very good, and my comments are quite minor.

The abstract could perhaps mention the high levels of heterogeneity observed in the meta-analyses.

The confidence interval reported on line 438 (0.07; 0.01-.03) is clearly wrong.

I find it more useful for funnel plots to have the funnel added.

[LINK]

---

## [Decision Letter · Decision Letter 1]

25 Feb 2020

Dear Dr. Jarvis,

Thank you very much for re-submitting your manuscript "Metabolic risk factors and incident advanced liver disease in NAFLD: A systematic review and meta-analysis of population based observational studies" (PMEDICINE-D-19-04140R1) for consideration at PLOS Medicine.

I have discussed the paper with editorial colleagues and our academic editor, and it was also seen again by two reviewers. I am pleased to tell you that, provided the remaining editorial and production issues are dealt with, we expect to be able to accept the paper for publication in the journal.

[LINK]

Please let me know if you have any questions. Otherwise, we look forward to receiving the revised manuscript shortly. 

Sincerely,

Richard Turner PhD, for Louise Gaynor-Brook, MBBS PhD

Associate Editor, PLOS Medicine

rturner@plos.org

Requests from Editors:

Please spell out "NAFLD" in the title. 

Around line 70, prior to the sentence summarizing study limitations, we ask you to add an additional sentence summarizing the findings for other metabolic risk factors, similar to the sentence at line 456. 

Please mention the issue of possible publication bias in your abstract. This could be quoted as a limitation, for example, or you could add a sentence just prior to the sentence on limitations to mention the findings quoted around line 347. 

Please subdivide the "author summary" into three sections (i.e., why was the study done/what did the researchers do and find/what do these findings mean), and adjust the content to ensure that each subsection includes about 3 points of 1-2 short sentences each. 

Please refer to the attached PRISMA checklist around line 180.

In table 1, the second column of the row for the Simeone 2017 study contains a question mark. Please complete this cell if there is a number missing. 

At line 376 and all other instances, please amend the text to "p=0.026" (or as appropriate). 

Around line 520, please cite one or two of the known limitations of analyses of observational data, e.g., the question of unmeasured confounding. Also, please mention the issue of possible publication bias. 

Could a relative lack of data from developing countries be seen as a limitation?

Please remove all instances of "[Internet]" from the reference list.

Noting reference 12, please ensure that all references have an individual or institutional author listed. 

Please add full access details for references 22, 31, 48-50, 52 and 55. 

In your figures, please quote exact p values or p<0.001, unless there is a specific statistlcal justification for reporting lower p values. 

Comments from Reviewers:

*** Reviewer #1: 

Thank you for the clarifications and additions. I have no other comments.

*** Reviewer #3: 

Alex McConnachie, Statistical Review

The authors have addressed the few comments I had, and I have no further observations. I think the statistical parts of the paper are very strong.

***

[LINK]

---

## [Editor Report · Decision Letter 2]

6 Apr 2020

Dear Dr Jarvis, 

On behalf of my colleagues and the academic editor, Dr. Amit Singal, I am delighted to inform you that your manuscript entitled "Metabolic risk factors and incident advanced liver disease in non-alcoholic fatty liver disease (NAFLD): A systematic review and meta-analysis of population based observational studies" (PMEDICINE-D-19-04140R2) has been accepted for publication in PLOS Medicine. 

PRODUCTION PROCESS

PRESS

PROFILE INFORMATION

Thank you again for submitting the manuscript to PLOS Medicine. We look forward to publishing it. 

Best wishes, 

Louise Gaynor-Brook, MBBS PhD

Associate Editor 

PLOS Medicine

plosmedicine.org